# Rural Morphology and Forces Driving Change in Rapidly Urbanizing Areas: A Case Study in Fujian, China

**DOI:** 10.3390/ijerph18094590

**Published:** 2021-04-26

**Authors:** Lishan Xiao, Peiqi Shi, Tong Lin, Ning Chen, Sha Huang

**Affiliations:** 1School of Environmental and Geographical Sciences, Shanghai Normal University, Shanghai 200234, China; lsxiao@shnu.edu.cn (L.X.); shipqxzz@163.com (P.S.); lintong128@163.com (T.L.); chenning010426@163.com (N.C.); 2Institute of Urban Study, Shanghai Normal University, Shanghai 200234, China

**Keywords:** rural settlement, urbanization, fractal dimension, rural transformation, rural planning

## Abstract

Rapid urbanization in China has transformed many rural areas from agriculture-dominated to diverse systems, but studies of rural morphology are limited compared to studies of urban settlement. This paper uses a fractal dimension (FD) value to analyze the change in rural morphology in Fujian Province, a region with a long history of rural settlement and rapid recent urbanization, and to explore the factors that influenced this change. We found that the rural FD value increased from 2000 to 2012 and that rural morphology was spatially heterogeneous. FD was generally lower than in urban areas but very close to a typical urban area value in the southeast coastal region. A structural equation model was used to identify key factors influencing rural morphology, which were natural conditions, rurality and economic development, while historic administration had the smallest positive effect. With a long history and unique administrative system, the spatial morphology of Chinese rural areas has shown characteristics distinct from compact urban or scattered rural areas. The urban planning method adopted by rural planners is not suitable in rural regions, because the planning potential of rural areas with high and low FD values varies. Although rural planning currently uses a very similar approach to urban planning, it should use a local, flexible and adaptive policy based on rural morphological characteristics.

## 1. Introduction

Urbanization influences the surrounding rural area in terms of economic structure, employment and lifestyles, and so even the remotest villages may be transformed into increasingly complex and heterogeneous shapes [1]. Modern urban–rural areas form a complex web of connections for resources, energy, information and population [2,3,4], and as a result of the interactions between rural areas and the surrounding environment, the rural system may either grow, decline or even vanish [5]. In many rural areas in China, agriculture is no longer the dominant economic sector, and households typically rely on a combination of agricultural and nonagricultural income for their livelihood. In more traditional rural areas, arable land loss, population ageing and outmigration have led to a rural decline [6], evidenced by decreasing employment, depopulation and economic stagnation.

Since 2011, more than half of the population of China has been living in urban areas [7]. However, there were still 564 million rural residents at the end of 2018, accounting for 40.4% of the total population. The contribution of agriculture to economic output declined from 50.5% in 1952 to 7.2% in 2018, and rural income per capita is only about 37% of urban income. Chinese planning policy has prioritized urban development and sacrificed rural interests for a long time, causing many rural–urban conflicts [8], and the existing framework in China for rural development is mainly composed of one-size-fits-all policies [9]. The urban planning method adopted by rural planners is not suitable in rural regions. China’s urban planning is a top-down process. The current rural planning procedures attach importance to centralized spatial reconstruction with intensive land use. The planning time is very short within half a year, and planners cannot conduct surveys in rural areas. Due to the lack of knowledge of agricultural production and demands, as well as understanding of the distribution and evolution of rural settlements, rural planners tend to copy the urban planning model without considering the incentives of rural residents in the short term. The large-scale demolition and reconstruction of houses have become the main content of rural planning, and this typically neglects the disparity between different regions and residents’ demands, and conflicts with the production and lifestyle of rural residents [10,11]. Rural planning using urban intensive spatial planning and centralized spatial reconstruction methods destroys the original landscape of the countryside [12] and may also lead to environmental problems caused by the compact development of cities, resulting in local resistance or even social conflict [13,14]. In this sense, rural planning is difficult to implement. In recognition of the problems caused by relative rural decline, the sustainable development of rural areas is now a government priority, and a “rural vitalization” strategy aimed at realizing the comprehensive rejuvenation of rural industry, culture, ecology, governance and human resources was put forward in 2019. However, there remains a possibility that rural revitalization planning may fail to appreciate the diversity and complexity of rural systems.

China’s administration system may result in the emergence of a unique spatial morphology with characteristics distinct from traditional urban and rural areas. In contrast to the definition of rural area used by the Organization for Economic Co-operation and Development (OECD, Paris, France) [15] or USA [16], Chinese rural areas are divided based on administrative divisions, which also form the basis of the residential registration system (“Hukou”). People who have a rural Hukou and an allocation of a homestead and agricultural land cannot also freely access urban soft infrastructure, such as education and housing subsidy, due to Hukou restrictions. Rapid urbanization combined with the registration system has dual effects on the rural system, simultaneously accelerating nonagricultural rural economic development (e.g., agricultural tourism or township enterprises) in some areas, while other areas become depopulated, and agricultural land is abandoned. The interplay among agricultural livelihoods, rapid urbanization, the administration system and historical geography complicates the rural morphology. 

Urban morphology is the study of the formation of human settlements and the process of their formation and transformation. This study seeks to understand the spatial structure and character by examining the patterns of its component parts [17]. Urban spatial patterns have already been quantified as they are considered to be closely related to urban sustainability. Although urban morphology has become more compact during the urbanization process in general, there are different spatial characteristics in mega, medium-size and small cities [18]. Compact urban morphology is helpful to develop efficient public transportation [19] and reduce commuting times [20], and also has potential benefits for personal relationships and physical health [21]. On the other hand, compactness may also hamper access to greenspace and water resources [22], increase household energy consumption along with air and noise pollution [23] and increase anxiety and stress [21]. A better understanding of urban morphology provides a reference for urban planning and increases sustainability [24]. However, compared to urban systems, rural spatial patterns have not been addressed in sufficient detail. The rural system was assumed to be self-sufficient in traditional Chinese society; however, it currently relies more on external resource supplies [4] and often suffers from serious pollution when urbanization occurs [25]. There is an urgent need to understand how rural spatial patterns evolve during the urbanization process to avoid environmental problems faced by cities today and to provide a basis for scientific rural planning in China.

The key objectives of this study were to ascertain: (1) how rural morphology changes during the urbanization process, and (2) how the factors combining modern urbanization and natural conditions have influenced rural morphology in areas with a long history of human settlement. We took the rapid urbanization of the south-eastern coast of China in Fujian province as an example, and used a fractal dimension (FD) methodology to assess changes in rural settlement morphology between 2000 and 2012. We established a structural equation model (SEM) to evaluate the influence of various factors and multiple causal pathways among variables. The paper helps to explain changes in rural morphology and provide implications for rural planning to increase rural sustainability.

## 2. Literature Review

### 2.1. Rural Morphology

A rapid and automatic calculation of urban morphology metrics includes urban patches, mean patch area and eccentricity of the standard ellipse [26]. Corresponding to urban expansion, researchers have used compactness indices to describe urban morphology based on the shape of regions by remote sensing data [27]; socioeconomic activity density by statistics data [28]; degree of mixed land use by using big data such as point of interests (POIs) [29]. The lack of social and economic statistics in rural areas restricts the application of socioeconomic density. POIs are not commonly distributed in rural areas, which hinders the new data application.

Urban morphology has no characteristic scales and cannot be effectively described by conventional mathematical methods, such as land area and perimeter [30]. Fractals describe complex geometric phenomena and reflect the degree of space filling; in other words, fractal change can reflect urban growth. The fractal dimension (FD) is an important index of a fractal. FD is a standardized value ranging between 0 and 2 to describe urban morphology objectively and comparably [24,30,31]. FD has been widely used in qualified urban morphology. FD is especially valuable for making comparisons and modeling urban growth. Fractal methodology can be used to describe the dynamic spatial patterns and processes that underlie socioeconomic interactions [32]. FD has been used to understand deforestation [33], urban growth [34,35,36]), environmental quality change [37], carbon footprint and road network [38] and urban microclimate [39] during the urbanization process. Rural settlements are an important component of rural areas [40]. They contain dwellings and facilities, thus making them an excellent measure of rural land subjected to nonagricultural use [41]. The FD of a rural settlement has the ability to show complexity and heterogeneity along the urban–rural gradient. This paper attempts to compare the differences between urban and rural morphology, and to explore the changing trends of rural morphology in the process of urbanization. The FD can meet these two requirements. In this paper, it is used to quantify rural settlement morphology under rapid urbanization.

### 2.2. Drivers of Rural Settlements

The distribution of rural settlements has a close but nonlinear relationship with natural conditions, demographic factors and economic factors. Natural conditions are the most important factor, among which elevation is most commonly considered to be a basic natural environmental factor that affects the distribution of rural settlements [42,43,44]. Luo et al. [45] found that nearly 80% of rural settlements were gathered with a slope of 6~25°, while the elevation had a higher significant agglomeration effect on rural settlements than slope. Precipitation and temperature were positively related in arid regions, while there was no apparent correlation in Southeastern China [46]. Climate-related disaster reduced land area and then influenced the rural poor population’s livelihoods significantly in a short time [47]. The increase in rural settlements is directly related to the loss of cultivated land [41]. The rural population slightly increased along with the settlement area before 2000 in China, and then it declined, while the rural settlement area continues to grow [41]. In economically developed areas, rural areas rely on market centers, resulting in large-scale residential communities formed in scattered rural residential areas and leading to changes in urban–rural relations [48]. The strength of these influencing factors varies in different regions, for example, less-developed rural settlements are more likely to be influenced by natural disasters or geophysical conditions, while developed rural areas are more likely to be influenced by urbanization. China has a long history of traditional agriculture, and cultural heritage plays an important role in the emergence of rural settlements, but the historical factor has received relatively little attention, and the effects of historic factors on maintaining rural settlements has not been quantified [48,49].

In addition, very little research has been conducted to quantity the driving forces of rural settlements and compare the importance of each potential influencing factor. Urbanization may create an imprint in the rural morphology, and the evolution of rural morphology is a confluence of modern urbanization and historical characteristics. Understanding the importance of each factor in explaining rural morphology can support future urban planning and enhance the sustainable urbanization process in China.

## 3. Research Area and Method

### 3.1. Research Area

Fujian is located in coastal Southeast China. The provincial government’s Functional Zoning Planning divides the province into optimizing development, priority development, restricted development and prohibited development zones according to National Functional Zoning Planning [50]. Optimizing development and priority development zones have a higher Gross Domestic Product (GDP) per capita and dense populations. Restricted development areas are used to provide food and other ecosystem services, and are, therefore, not suitable for intense exploitation and industrialization. The prohibited development zone represents nature reserves, cultural and natural heritage, scenic spots, forest parks and geological parks. Each county is designated as either as an optimizing, priority or restricted development zone, while prohibited development zones are decided independently of county boundaries (Figure 1).

Fujian is a province with a long history. Demographic information has been recorded since the Tang dynasty (in the year of 618 AD) and shows that the population has often fluctuated but has generally increased. The population decreased in the first half of the 20th century, but since the establishment of the People’s Republic in 1949, the population has grown rapidly (Figure 2).

Fujian Province is a rapidly urbanizing area. According to calculations by the National Bureau of Statistics, the urbanization rate is the ratio of the urban permanent population to the total population. In the beginning of China’s economic reforms and opening up in 1978, the urbanization rate was 13.7%, and in 1990, 2000 and 2012, it was 16.7%, 41.6% and 59.7%, respectively. The national urbanization rate in the corresponding period was 17.9%, 26.4%, 36% and 51.3%. The urbanization growth rate of Fujian Province is higher than the national average. Fujian has undergone extremely rapid economic development since 1978, with the GDP per capita in 2012 reaching USD 5886. about 33.5% higher than the national average, and an urbanization rate of 60.8%, 13.9% higher than the national average. The contribution of agricultural output to GDP dropped from 36.0% in 1978 to 17.0% in 2000 and 9.3% in 2012, while the percentage of the population directly dependent on agriculture was 75.1%, 46.8% and 29.2% in these three years, respectively. Rural income per capita in Fujian was only US Dollars (USD) 1070 in 2012, and income from agriculture accounted for 34.4% of the total income of rural residents, a reduction of 6.4% since 2000. Income from the industrial and service sectors accounted for 41.7% of total income, an increase of 26.0% between 2000 and 2012.

### 3.2. Research Method

#### 3.2.1. Fractal Dimension

The box-counting approach is a bottom-up method that is frequently used to estimate fractal dimension [36,54]. In this study, FD was computed using a box-counting algorithm [55] where the calculated area was occupied by a rectangle with length (L). If the area is covered by one box, l = L, the box number with land cover is *N*(L) = 1; if the length is divided by two parts, l = L/2, the total box number is 4, and the box number with land cover is *N*(L/2^2^). For each iteration number r, the length was divided into equal parts with the length of l = L/2^r^, the total number of box is 2^r + 1^, and the box number with land cover is *N*(L/2^r^). The calculation was iterated 9 times. The box-counting fractal dimension was the coefficient obtained by regressing the dependent variables as Equation (1):(1)$$\log N\left(\frac{L}{2^{n}}\right)=-FD\log2^{n}+C$$
where *FD* is the fractal dimension, and *C* is a constant. *FD* ranged from 0 to 2, where *FD* = 2 means that the occupied area was distributed uniformly or in perfect homogeneity and *FD* = 1 is the threshold in fractals [1]. Lower FD typically represented areas with scattered settlements, whereas fractal dimensions close to 2 are associated with compact built-up areas [36]. The quality of fractal analysis relies on the *R^2^* of the linear regression, where higher *R^2^* values indicate a more robust fractal property.

#### 3.2.2. Structural Equation Model

A structural equation model (SEM) based on a cross-sectional dataset was used to analyze the factors influencing FD. SEM is a statistical technique for testing and estimating causal relations which uses a combination of statistical data and qualitative causal assumptions. SEM integrates factor analysis and path analysis, and examines the causal relationships among a set of variables within an integrated framework [56]. SEMs are increasingly used in ecological and geographical research as a multivariate analysis that can represent theoretical variables and address complex sets of hypotheses. The database was created in SPSS (IBM Corp. Armonk, NY, USA) and then imported into AMOS 18.0 (IBM Corp. Armonk, NY, USA). Various measures exist to assess the goodness-of-fit of a SEM, including chi-square (*χ^2^*), goodness of fit index (GFI), comparative fit index (CFI), normed fit index (NFI) and Akaike information criterion (AIC).

#### 3.2.3. Influencing Factors

We hypothesized that natural conditions, urbanization level, socioeconomic status and historic administration were potential factors that influenced rural settlement morphology. Elevation above sea level was the best available measure of natural conditions, and agricultural output value and rural income were used as indices for economic factors. The numbers of towns and villages were used to represent administrative factors, and also encompass historical geographical information. Towns and villages are the basic administrative units in China, and administrative divisions were formed by the interaction of economic development and historical geography. After more than 1000 years of development, Chinese society has formed a relatively complete and mature administrative division system [57,58] which is characterized by path dependence [59], and there is significant inertia in the administrative system when responding to changes in the location of major economic centers. To facilitate efficient management over time, more administrative divisions have been formed as local societies developed and populations grew, and in general, the faster the economy developed, the more administrative divisions were set up [60]. Fujian province had 9 prefecture-level cities, 84 counties, 1104 towns and 14,335 villages in 2012.

The variable used to describe urbanization was rurality, which was first introduced by Cloke [61,62] to distinguish urban and rural characteristics in Britain. The concept of rurality reflects a more comprehensive urbanization level than that presented in government statistics, and rurality index analysis has significantly improved the understanding of the recent development of rural China [9]. Our calculation was based on Waldorf’s definition [63], which includes four dimensions in the rurality index: population size, population density, extent of urban (built-up) area and remoteness. The index varies from 0 (lowest rurality, most urban) to 1 (highest rurality, most rural). The detailed calculation is shown in Equation (2):(2)$${\mathrm{IRR}}_{i}=\frac{1}{4}\left[\frac{{\left(\mathrm{lg}PS\right)}_{max}-{\left(\mathrm{lg}PS\right)}_{i}}{{\left(\mathrm{lg}PS\right)}_{max}-{\left(\mathrm{lg}PS\right)}_{min}}+\frac{{\left(\mathrm{lg}PD\right)}_{max}-{\left(\mathrm{lg}PD\right)}_{i}}{{\left(\mathrm{lg}PD\right)}_{max}-{\left(\mathrm{lg}PD\right)}_{min}}+\frac{{\left(\mathrm{lg}B\right)}_{imax}-{\left(\mathrm{lg}B\right)}_{i}}{{\left(\mathrm{lg}B\right)}_{imax}-{\left(\mathrm{lg}B\right)}_{i}}+\frac{{\left(\mathrm{lg}D\right)}_{imax}-{\left(\mathrm{lg}D\right)}_{i}}{{\left(\mathrm{lg}D\right)}_{imax}-{\left(\mathrm{lg}D\right)}_{i}}\right]$$
where max represents the maximum value; min represents the minimum value; *i* is the variables included in Equation (2), including population size, population density, extent of urban (built-up) area and remoteness, respectively. IRR*_i_* is the index of relative rurality, *PS_i_* is population size, *PD_i_* is population density, B*_i_* is built-up area and *D_i_* is remoteness (the distance between area *i* and the nearest downtown area). *PS_i_, PD_i_* and *B_i_* are positive indicators, and *D_i_* is a negative indicator. The higher the IRR, the lower the urbanization level, while lower values indicate a greater urbanization influence. The statistics of *PS*, *PD*, *B* and *D* are listed in Table 1.

#### 3.2.4. Data Source

The land use data with a 30 m resolution for 2000 and 2012 were based on Landsat TM images and were provided by the Data Center for Resources and Environmental Sciences of the Chinese Academy of Science. The social and economic data are derived from the Fujian Provincial Statistical Yearbook.

## 4. Results and Discussion

### 4.1. Changes in Rural Settlement Area

Rural settlement areas in the eastern coastal area were larger than those in the western mountainous area (Figure 3). For a long time, the coastal area and the northwest interior were isolated from each other by mountainous terrain which made communication difficult [60], so the distribution of rural settlements remains uneven.

Rural settlements witnessed dramatic changes as urbanization accelerated between 2000 and 2012. Despite the decreasing rural population, the area covered by rural settlement increased by approximately 39.85%, from 1212 km^2^ to 1695 km^2^. However, the increase in settlement areas in restricted zones was 50.7% compared to 35.6% in optimizing and priority development zones. Only five of the 84 counties saw decreases in the rural settlement area. Urban sprawl in two of these counties has led to many rural settlements being formally incorporated into larger urban centers (“changing village into urban community”), and the other three counties were located in the inland mountainous area of the restricted development zone (e.g., in Zherong County, 94.6% of the total area is classified as mountainous terrain). In remote mountain villages, there is relatively little cultivated land and few employment opportunities, forcing people to migrate to urban areas.

The 15 counties that experienced the most rapid growth in rural settlement area were all located in the restricted development zone. The largest increases in rural settlement areas were in Pingnan and Shouning counties (by a factor of 4.03 and 5.23, respectively). The area of rural settlement in the restricted development zone increased faster than in the optimizing and priority development zones, but this increase was accompanied by a contrasting population decline, a fact inconsistent with the provincial government’s Functional Zoning Plan. One possible explanation is that when rural residents migrate to urban centers, they lack security and a sense of belonging (especially if they are unable to obtain an urban Hukou) and so are reluctant to give up their rural land endowed by the rural registration system, which is left uncultivated, while their savings are used to build a bigger house in the village [64,65]. Yu et al. similarly proposed that residents in mountainous villages choose to move to central town-villages rather than cities in search of better public services [66]. However, they may still return to the village to engage in agricultural production and often have the ability and willingness to enlarge their houses.

### 4.2. Changes in Rural Morphology

A comparison of the fractal dimension between 2000 and 2012 is shown in Figure 3 and Table 2. The fragmentation of rural settlement was reduced, and the quality of the fractal (*R^2^*) increased. In addition, FD increased going from the western mountainous region to the eastern coastal region, and significant spatial heterogeneity was observed between northern and southern regions (Figure 4). Similar to those of urban systems [67,68], features of self-similarity were observed in rural regions with an FD value larger than 1.

The FD of remote villages in the north and west was smaller than 1.0, which meant that the rural settlements were mostly isolated and fragmented, while in the more developed southeastern coastal region, the FD was greater than 1.44. The average FD of cities worldwide was found by Encarnacao et al. to be 1.7, and to generally increase over time [36]. In Wallonia in Belgium, FD values were greater than 1.4 [1], while in 42 metropolitan regions of the United States of America (USA), the FD ranged from 1.708 to 1.960. These results show that FD values for Fujian rural settlements were generally lower than those for urban areas. However, in eastern coastal areas, the FD values were not statistically different and approximated those of a typical urban area. Pingnan County in the north of Fujian had the lowest FD value in both 2000 (*D* = 0.56, *R^2^* = 0.9300) and 2012 (*D* = 0.8, *R^2^* = 0.9772), and the quality of regression was the third lowest in the province; the urbanization rate was 37.9%. In 2012, rural income in Pingnan County was 60.2% below the average for Fujian Province, and rural settlements were small and isolated, but the FD value had increased by 45% since 2000. The county with the highest FD value was Jinjiang County in the southeast of the province (in 2000, *D* = 1.59, *R^2^* = 0.9943; in 2012, *FD* = 1.60, *R^2^* = 0.9967). As this area is highly urbanized not much undeveloped land remains, and so the FD value remained relatively stable (increasing by only 4% during the study period). For decades, Jinjiang County has been among the top ten counties in terms of economic performance in China, and is the county with highest income per capita in Fujian Province; it is famous for its rural township enterprises, especially in manufacturing. Rural income per capita was USD 1548, and living area was 65.66 m^2^ per capita in 2012, which was 44.7% higher than the average for Fujian Province. In terms of FD, the rural settlement distribution in Jinjiang approached that of a typical urban area and became more compact between 2000 and 2012. Figure 5 illustrates the different landscapes of Pingnan and Jinjiang counties.

Blue triangles in the left represent the restricted development region and red dots the optimizing development and priority development zones. Figure 5a,b are log–log plots of the fractal dimensions of rural settlements in Jinjiang and Pingnan counties, with the highest and lowest FD values. Red asterisks indicate the number of corresponding iterations in 2000; blue circles indicate the number of corresponding iterations in 2012. Pictures in the left corner of Figure 5b,c show rural settlements in Jinjiang and Pingnan counties.

In 2012, the average FD value for restricted development regions was 1.03, an increase of 6.5% from 2000. Optimizing and priority development zones had an average FD of 1.29, a 3.7% increase from 2000. The ten counties with the lowest FD values in 2012 were all located in restricted development regions, while the ten counties with the highest FD values (*FD* > 1.45) were located in optimizing and priority development zones. The FD values in restricted development regions increased dramatically over the course of the study compared to other regions, and these trends were similar in rural settlement areas. This finding is inconsistent with the Functional Zoning Plan, a fact that should be taken seriously in view of the potential threat to food supply and ecosystem services.

### 4.3. Factors That Influence the FD

We hypothesized that agricultural output value, rural income, elevation, rurality, and the numbers of towns and villages were factors that would influence rural settlement distribution in terms of FD (Figure 6). However, agricultural output was found to be only loosely related to FD and was, therefore, eliminated from the model; subsequent indices then performed better (*χ*^2^ = 75.28, *p* = 0.284 > 0.05, goodness of fit index (GFI) = 0.948 > 0.9, CFI = 0.936 > 0.9, root mean square error of approximation (RMSEA) = 0.023 < 0.05). The final model and standardized effects are depicted in Figure 7. The path analysis illustrated that elevation and rurality had negative effects on FD, while rural income and numbers of towns and villages had positive effects. In terms of the absolute values of standard path coefficients, the order of effect was elevation, rurality, rural income and numbers of towns and villages.

Solid lines indicate positive effects, dashed lines indicate negative effects, and the width of the line indicates the magnitude of the effect.

#### 4.3.1. Elevation

The variable with the largest effect on rural settlement was elevation, which has provided a basis for rural settlement sites and layout since ancient times and evidently continues to influence development today. The terrain of Fujian is high in the west but low in the east, and mountains and hills cover about 80% of the land area. The distribution of rivers is consistent with the mountains, and ancient rural residents chose to settle near fresh water. Rural settlements developed along the valleys and, therefore, present elongated or tentacular characteristics. The terrain slopes gently along the east coast (gradient generally between 0 and 10 degrees; elevation 0–200 m), and this is where most rural inhabitants preferred to settle in concentrated areas. The average patch size of these villages was larger than in inland regions, and the degree of landscape fragmentation was also less in coastal regions. As elevation increased, the density of rural settlements decreased, as did the FD value (*r* = −0.659, *p* = 0.000).

#### 4.3.2. Rurality

Rurality, an index of the level of urbanization, ranged between 0.13 and 0.96, with an average of 0.65 in 2012 (Figure 6). Rurality declined from 0.66 in 2000 to 0.64 in 2012. Across Fujian Province, rurality increased from southeast to northwest in both 2000 and 2012, and was correlated negatively with FD (*r* = −0.683, *p* = 0.000). Rurality had a negative effect on FD, 0.351.

#### 4.3.3. Rural Income

Rural income was positively related to the FD value (*r* = 0.492) at the α = 0.01 level. In rural areas, the nonagricultural economy was as important as it was in urban areas. As evidenced by Liu and Liu [69], the proportion of conventional on-farm households decreased, while off-farm households increased (and which gained a living from, e.g., rural tourism and manufacturing), so rural income has gradually decoupled from agriculture. This could explain why the value of agricultural output did not directly correlate with FD in the SEM model.

#### 4.3.4. Number of Villages and Towns

FD correlated significantly with the numbers of towns and villages (*r* = 0.205, *p* = 0.030). The number of towns and villages is higher in the southeast, leading to higher FD values. Rural settlements are inherited from ancient administrative boundaries, but because the economic center has changed gradually from the northwest of the province to the southeast coastal area today, the numbers of towns and villages showed the least propensity to influence rural settlement morphology compared to other variables, with an effect of 0.193.

### 4.4. Implications for Rural Planning

In the SEM model, the number of towns and villages and elevation will remain stable in the long run, while increasing rural income and decreasing rurality are likely to continue to increase the rural FD value in future. Increasingly compact rural development is likely to cause some environment problems. The results indicated that many rural settlements in Fujian are neither urban nor rural but demonstrate some features of both types, and bear the marks of both history and modern urbanization.

#### 4.4.1. Rural Areas with High FD Values

Rural area with high FD indicated that a new pattern of settlement transition is taking shape. McGee [70] labeled these new dynamic patterns with the Indonesian term ‘‘desakota’’, which combines the words for village (desa) and town (kota), as a new kind of tight mosaic of urban and agricultural land uses in the most populous and rapidly developing regions. Zhao et al. [71] proposed the term “town village” to describe rural urbanization in China under the Hukou registration system. Population is a very important factor considered in planning. The paper shows that due to the residential registration system (“Hukou”) system implemented in rural areas, people living in these places are classified as rural populations. However, these areas where they live are already close to the city in terms of FD, and the planning methods need to be distinguished from scattered rural areas. Increasing FD correlates with a rising spatial efficiency of land use (for example, in compact development). This also correlates, however, with a reduction in cultivated land and forests in rural regions, and a reduced connectivity between built-up regions and ecological land. For rural areas with higher FD values that are located in developed regions, integrated urban–rural developments should be encouraged. Higher FD values almost always lead to higher environmental emissions [34], and in this case, rural areas could possibly encounter the same environmental problems as typical urban areas. Environmental infrastructure, such as wastewater treatment and waste disposal facilities, should be built to cope with increasingly compact rural development, and greenspace should be preserved to avoid the heat island effect which is faced by compact cities.

#### 4.4.2. Rural Areas with Low FD Values

The SEM result indicated that rural morphology is still influenced by historical geographical factors. Rural areas with lower FD values should preserve more localized, isolated, and dispersed rural settlements, which could preserve important aesthetic, historical, and ecological values; this is the intention of the provincial Functional Zoning. In contrast to optimizing development zones, our results showed that restricted development zones have undergone the most rapid growth in rural settlement areas and FD values, which may pose a threat to maintaining ecosystem services.

The results showed that rural morphology is partly inherited from ancient administrative boundaries. Traditional rural landscapes formed a basis for Chinese societies over centuries, but rural morphology indicates that many features from a remote past evolve much faster during urbanization. The processes and management of past traditional landscapes offer valuable lessons for the sustainable planning and management of future landscapes. People still occasionally return to nearly deserted villages to grow crops while usually living in central town-villages [56]. Rural planning should aim to provide better agricultural production and living infrastructure which complies with traditional and natural patterns to increase agricultural productivity, as this would help to attract these migrants back to agricultural production, preserve historical value and increase their living standards. These areas with scattered residential settlements can also serve as a reservoir of biodiversity to preserve ecosystem services.

## 5. Conclusions

Rural urbanization is a complex process to which various factors contribute. This paper focused on the distribution of rural settlement using FD and addressed the causal relationship between FD and various factors in a rapidly urbanizing Chinese province; it extends the understanding of changes in rural morphology in the context of urbanization and provides insights for rural planning policy.

Despite the decreasing rural population, the area covered by rural settlement increased by approximately 39.85% during the 12-year study period. Fractal analysis showed that FD increased during the decade and that the quality of fractal analysis also increased. The FD showed spatial heterogeneity, which was generally lower in rural regions than in urban regions, but very close to that of a typical urban region in the southeast coastal area, which became more compact over the course of our study. Urbanization changes the rural landscape and, in some cases, blurs the distinction between urban and rural regions. In view of the provincial Functional Zoning Plan and its rural protection strategy, rural settlement expansion and higher FD values should be discouraged in restricted development areas, but in fact, rural settlement areas and FD values are currently increasing faster than in optimizing and priority development zones. Rural morphology is a result of the long-term effects of multiple factors that relate not only to physical characteristics but also to human activities. Natural conditions, socioeconomic conditions, urbanization level and historic administration were all found to contribute to the distribution of rural settlement. Rural planning should take rural morphology and its changing trends into account, and adopt a local, flexible and adaptive method. In rural areas with high FD values, rural planning should aim to avoid typical urban environmental problems, including environmental pollution, resource overuse and heat island effects. In rural areas with low FD values, rural planning should aim to preserve historical, ecological and agricultural values. Fractal analysis has the potential to provide new insights into the complex urbanization process for rural areas. The SEM method is effective in addressing causal relationships and ranking influencing factors in terms of their importance. The methods used in our study can be used by rural planners to inform the changing rural morphology and its driving forces to enhance differential rural planning strategies for various urbanized rural morphologies.

## Figures and Tables

**Figure 1 ijerph-18-04590-f001:**
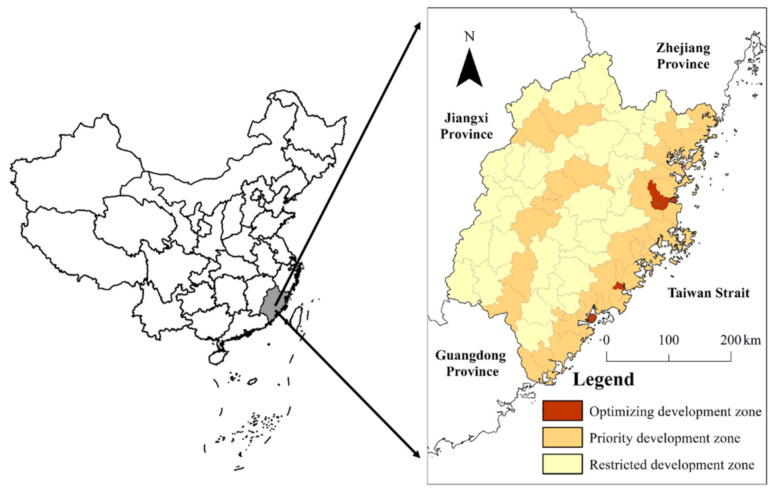
Location of Fujian Province and county-level functional zones.

**Figure 2 ijerph-18-04590-f002:**
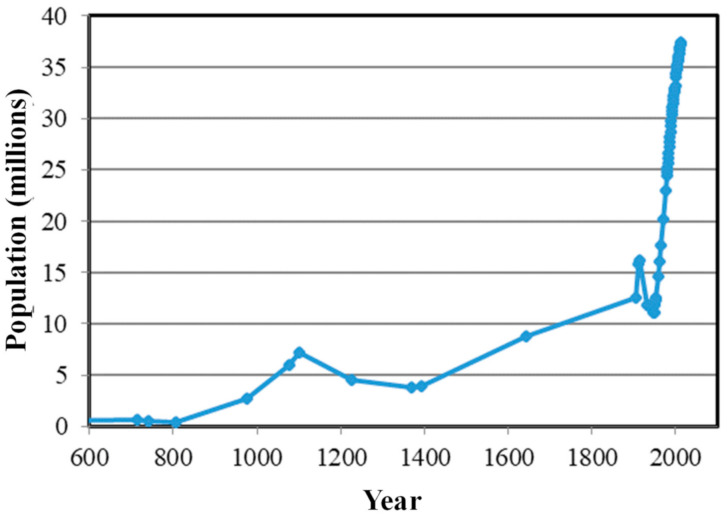
Population of Fujian Province in historical and modern times. Data sources: Xu; Ge; FSY [51,52,53].

**Figure 3 ijerph-18-04590-f003:**
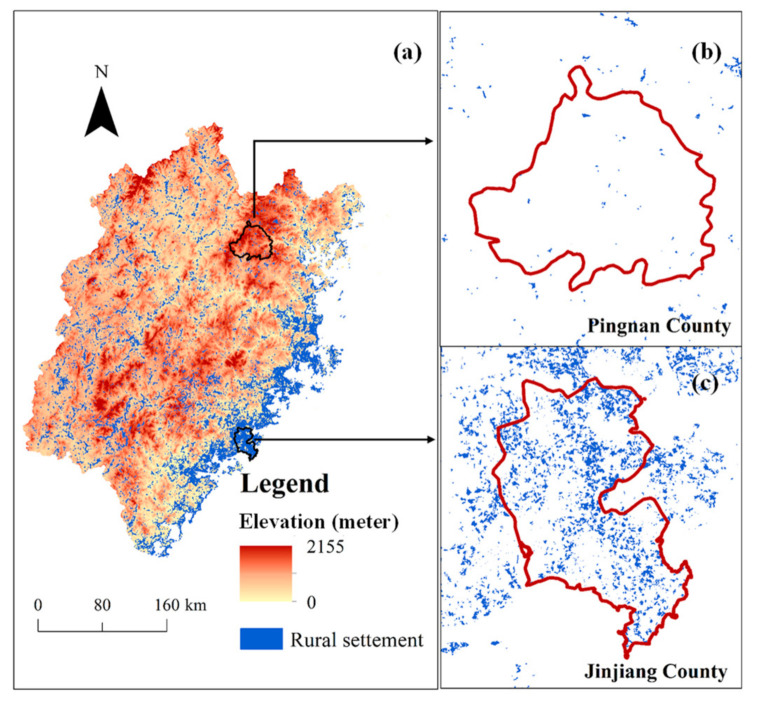
Distribution of rural settlements in (**a**) Fujian Province, (**b**) Pingnan County and (**c**) Jinjiang County.

**Figure 4 ijerph-18-04590-f004:**
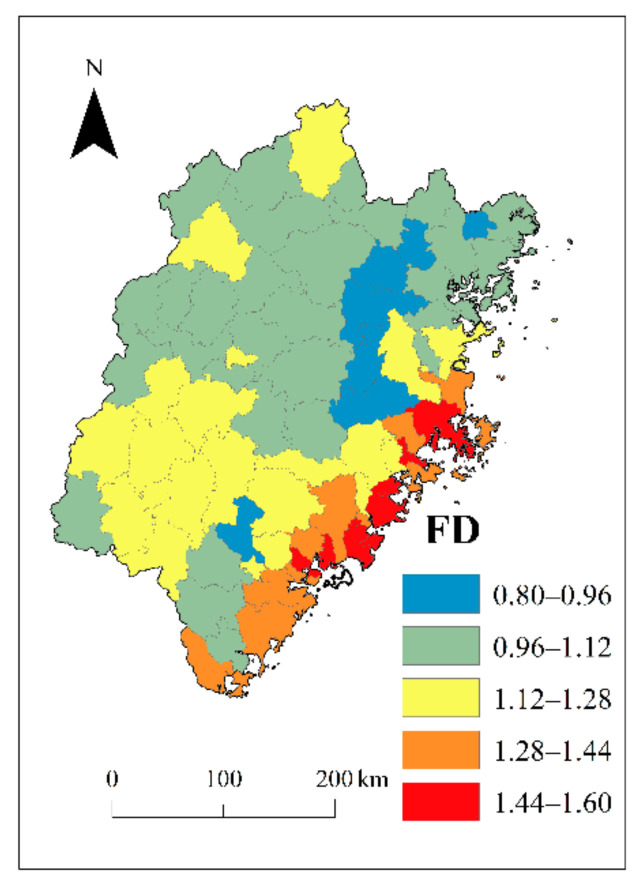
Fractal dimensions (FD) of Fujian Province in 2012.

**Figure 5 ijerph-18-04590-f005:**
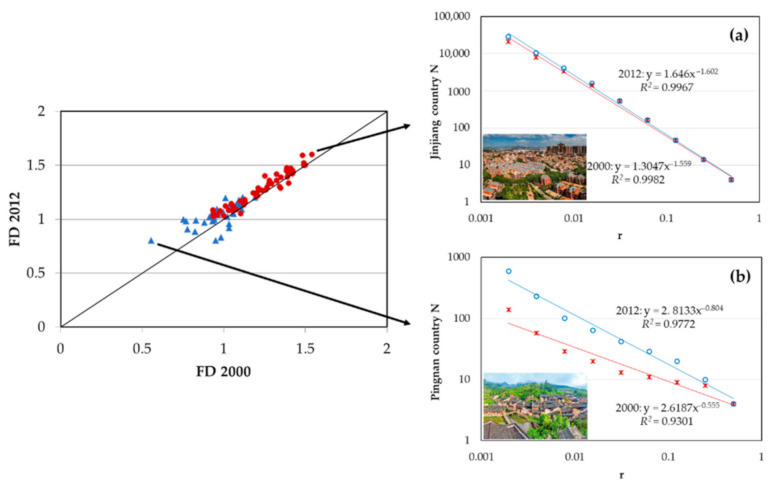
Fractal dimension values for 84 counties in 2000 and 2012; (**a**) log–log plots of the fractal dimensions of rural settlements in Jinjiang county; (**b**) log–log plots of the fractal dimensions of rural settlements in Pingnan county. FD: Fractal dimensions; x: 1/(2^r^); r: Iteration number; y = N: Box number with land cover.

**Figure 6 ijerph-18-04590-f006:**
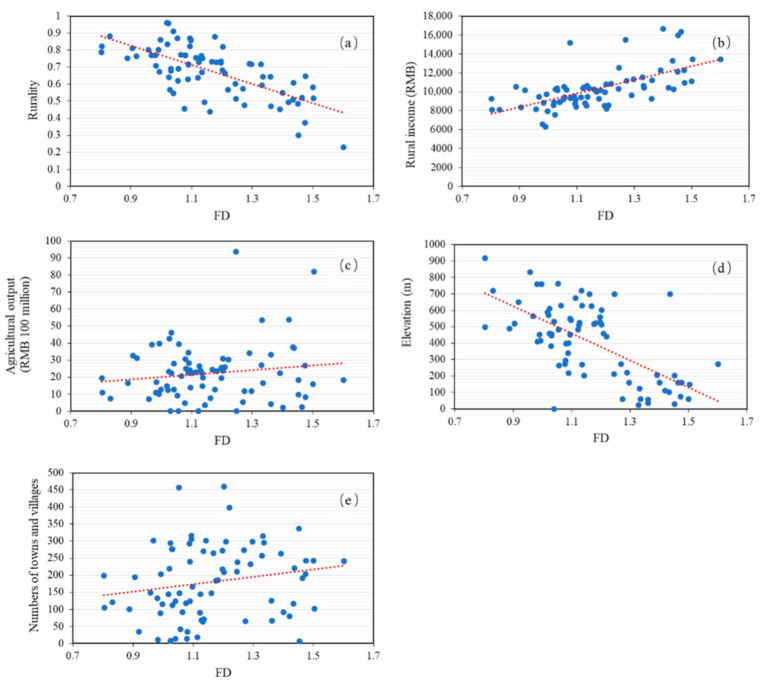
Hypothesized factors in the SEM model, including rurality (**a**), rural income (**b**), agricultural output (**c**), elevation (**d**) and the numbers of towns and villages (**e**). RMB: Renminbi; FD: Fractal dimensions.

**Figure 7 ijerph-18-04590-f007:**
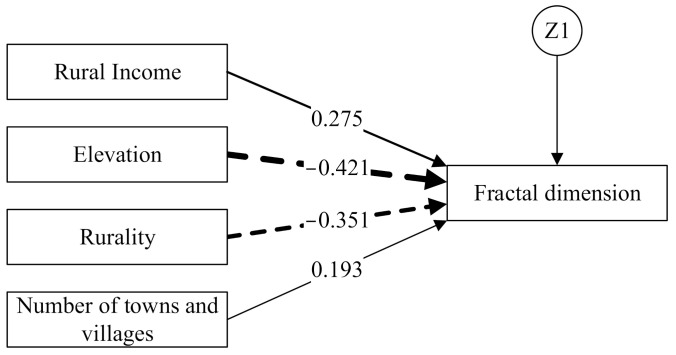
SEM model.

**Table 1 ijerph-18-04590-t001:** Variable statistics in the index of relative rurality.

	*PS*	*PD* Pop/km^2^	*B* km^2^	*D* km
Maximum	1,986,400	25,599	372	264
Minimum	83,951	59	4	0
Average	457,405	1781	64	49
Medium	349,550	316	47	42

*PS*: Population size; *PD*: Population density; *B*: Built-up area; *D*: Remoteness (the distance between area i and the nearest downtown area).

**Table 2 ijerph-18-04590-t002:** Characteristics of FD in Fujian Province.

Year	Variables	Average	*SD*	Skewness	Kurtosis	Max.	Min.
2000	FD	1.13	0.20	0.029	–0.372	1.54	0.56
	*R^2^*	0.9236	0.012	–3.530	15.152	0.9994	0.9236
2012	FD	1.18	0.19	0.293	–0.917	1.60	0.80
	*R^2^*	0.9910	0.008	–3.106	0.263	0.9997	0.9904

*SD*: Standard deviation; Max.: Maximal value; Min.: Minimum value; FD: Fractal dimensions; *R^2^*: Coefficient of determination.

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
