# Peer review of "Rural Morphology and Forces Driving Change in Rapidly Urbanizing Areas: A Case Study in Fujian, China"

_ijerph, 2021, doi:10.3390/ijerph18094590_

Round 1
Reviewer 1 Report
This paper uses FD indicator to analyze the characteristics of morphology change in the rural area of Fujian Province, aiming at revealing driving forces behind this phenomenon. The research results provide certain guidance for rural planning and its relevant policy-making.
The following questions need to be addressed:
1)Section 3: The authors claim that the study focuses on “rural areas in rapidly urbanizing areas”. It is necessary, therefore, to provide clear principles and methods to identity the rapid urbanization areas in Fujian.
2)Line50-52: the paper says that current planning practice has negative effects, it would be nice to discuss clear what planning concepts or methods cause this situation and how to improve.
3)Section 2.1: It is necessary to supplement the commonly used quantitative measurement indexes of morphology, as well as the reasons why the author chooses FD indicator and its advantages compared with other methods.
4)Line115-116: What is the relationship between them? Please add details, which will directly affect the selection of indicators in section 3.2.3 (influencing factors).
5)Section 3.2.3(influencing factors): Data source and basic information of the data need to be supplemented.
6)The selected underlying forces need more theoretic support, for instance, Line193-195. Why is the slope not selected? Terrain slope is also an important indicator, which will directly affect the possibility of housing construction.
7)Numbers of Some Sections are wrong, eg. 3.2.3(influencing factors).
Reviewer 2 Report
The link in reference 14 does not work (the requested resource could not be found).
The references [60], [61] and [34] in the text are joined with the previous or the following word.
Reviewer 3 Report
The authors use the term "village morphology", "urban morphology". I think that for better readability of the presented material it is necessary to explain what the author means or what definition he uses when using such a term.
The article indicates an important problem of Chinese spatial policy concerning both rural and urban areas. The specificity of the related problems is emphasized by the limitations of the system of assigning (recording) population to specific rural areas. I think that this aspect should be reflected in the recommendations related to the research results.
The authors point out important characteristics of rural areas, to the characteristics identified by the authors I would add the need for rural areas to serve as a reservoir of biodiversity.
In conclusion, a very interesting paper using advanced research methods such as SEM, definitely worth publishing.
Reviewer 4 Report
Line 30- it is not clear what “external environment” means
Line 39-40-40- Please, use the same places to the right of decimal point of percentage.
Line 61- please explain the meaning of OECD at least at the first time you use the abbreviation.
Line 133-142- In lines 133-134 seems that natural reservation are different from prohibited development zones while in line 139 is stated that prohibited development zones represents nature reservs,…Please explain better this point
Line 146: what “AD” means?
Line 150- “BC” in figure 2. Probably shall be deleted. Please express what x-axis represents.
Line 163- Research Method, a general consideration. -
In my opinion it would be useful to the understanding of the research paper to improve the “research method” section. For instance, the authors listed the influencing factors but no data were provided. In equation (2) what values are taken into account by the authors? Maybe a table with maximum and minimum values should help the reader to replicate the methodology proposed by the authors.
167- what’s the height of the rectangle?
Line 169- it’s not clear the equation: “l=rL=L/2r”
Line 216: it’s not clear “max” and “min” in equation (2) what refer to, the same for the index “i”.
Line 232-233- Why 2000 and 2012? What is the source of these data? It’s not very clear to me. Maybe it should be better explained in “research methods”
